# Unveiling the Essence of Poetry: Introducing a Comprehensive Dataset and Benchmark for Poem Summarization

**Ridwan Mahbub[1], Ifrad Towhid Khan[1], Samiha Shafiq Anuva[1],**
**Md Shihab Shahriar[1], Md. Tahmid Rahman Laskar[2,3],** and **Sabbir Ahmed[1]**
[1]Islamic University of Technology,      [2]York University,      [3]Dialpad Canada Inc.
{ridwanmahbub, ifradtowhid, samihashafiq, shihabshahriar}@iut-dhaka.edu
tahmid20@yorku.ca, sabbirahmed@iut-dhaka.edu

## Abstract

While research in natural language processing has progressed significantly in creative language generation, the question of whether language models can interpret the intended meaning of creative language largely remains unanswered. Poetry as a creative art form has existed for generations, and summarization of such content requires deciphering the figurative patterns to find out the actual intent and message of the poet. This task can provide the researchers an opportunity to evaluate the creative language interpretation capacity of the language models. Unlike typical text, summarization of poems is a challenging task as poems carry a deeper meaning, which can be easily lost if only the literal meaning is considered. That being said, we propose a new task in the field of natural language understanding called 'Poem Summarization'. As a starting, we propose the first-ever dataset for this task, named 'PoemSum', consisting of 3011 samples of poetry and its corresponding summarized interpretation in the English language. We have benchmarked the performance of different state-of-the-art summarization models and provided observations on their limitations. The dataset and all relevant code used in this work have been made publicly available[1].

## 1 Introduction

Poetry has been used for centuries as a powerful means of expressing the feelings, thoughts, ideas, and emotions of human beings (Freeman, 2000). It holds a profound significance in human culture and expression, transcending time and boundaries. Poem summarization refers to the process of reducing a poem to its core ideas to provide a concise and informative overview of the content (Wolosky, 2008). This not only gives us an idea about the main theme but also describes the style, rhyme, techniques, and metaphors used in the poem. Automatic summarization of such creative language can

make poetry more accessible to a wider range of audiences and applications (Upadhyay et al., 2022).

In recent years, there has been notable research conducted on text summarization in the field of Natural Language Processing (NLP) (Altmami and Menai, 2022; Widyassari et al., 2022; El-Kassas et al., 2021; Laskar et al., 2022). However, to the best of our knowledge, no such work has been done in the domain of poem summarization yet. While the summarization process of poems seems quite similar to the generic text summarization, there are some major differences between the two. Text summarization, in general, is a technique that reduces the number of sentences and words in a document by extracting its important information into a concise paragraph without changing its meaning (Erkan and Radev, 2004). Most of the summarization models being proposed in recent years have been trained on news-related datasets (Varab and Schluter, 2021; Hasan et al., 2021; Grusky et al., 2018; Laskar et al., 2020a,b), which mostly contain facts, so they do not have to look for deeper meaning while generating a summary. The same goes for datasets that are built using financial (Russell-Rose et al., 2002) or medical data (Gupta et al., 2021). This process only considers the literal meaning of words, and it does not focus on the deeper meanings. Applying this summarization technique directly to poems may not be helpful in capturing the essence. Hence, it is necessary to study if baseline NLP models (Raffel et al., 2020) are sufficient for poem summarization tasks.

Summarizing literary work poses lots of challenges. For instance, the artistic style of each writer is different, and the literature can be interpreted in many ways. Moreover, to understand the figurative language of poems, the model has to understand the interconnection between sentences and the deeper meaning conveyed by the poem as a whole. In order to facilitate research in this domain, we have introduced a new task in the field of natural lan-

---

[1] https://github.com/Ridwan230/PoemSum

guage understanding named 'Poem summarization' and proposed a novel poem summarization dataset. The main contributions of our paper are as follows:

- Introducing a novel poem summarization dataset named 'Poemsum'.
- Evaluating how the existing summarization models perform on this task and identified their limitations to facilitate future research.

## 2 Related Works

The text summarization task can be broadly divided into two categories: Extractive (Rau et al., 1989) and Abstractive (Chopra et al., 2016) methods. The extractive summarization approach highlights the key points in the source text, which are then taken out and compiled into a succinct summary. On the other hand, the abstractive approach uses a different collection of words compared to the original text to construct a paraphrase of the main ideas of a given text. For poem summarization, abstractive summary generation is required as only direct extraction of words from poems would not be enough to convey the actual meaning of the poem.

In the abstractive summarization domain, Narayan et al. (2018) introduced a dataset called 'X-Sum' by gathering 226,711 online articles from the British Broadcasting Corporation (BBC). In the same year, Grusky et al. (2018) proposed a summarization dataset named 'NEWSROOM' containing 1.3 million articles and summaries authored by writers and editors in the newsrooms of 38 major news publications that combine both abstractive and extractive summarization methods. The 'XL-Sum' dataset proposed by Hasan et al. (2021) covers 44 languages, and it is the largest abstractive summarization dataset in terms of the number of languages represented and the number of samples gathered from a single source. Although summarization has made great strides in recent years (Kumar et al., 2021), most works have used news articles as data source (Fabbri et al., 2019), where the focus is mostly on extracting key information, forming a shorter passage. Poems, on the one hand, do not benefit from such types of summarization, yet there are no studies aimed at exploring the prospects of poem summarization.

Apart from poetry, there has been a recent insurgence of datasets in other types of creative language (Mckeown, 2022; Chakrabarty et al., 2023a,b) with the objective of Natural Language Interpretation (NLI). For instance, Chakrabarty et al. (2022a) stud-

| Type | Size |
| --- | --- |
| Number of Poems | 3011 |
| Number of Poets | 930 |
| Max poem length[*] | 6830 |
| Max summary length[*] | 1104 |
| Avg. poem length[*] | 209 |
| Avg. summary length[*] | 141 |
| Avg. no. of poems per poet | 3.24 |

[*]Unit represented in number of words

Table 1: Statistics of the PoemSum Dataset

ied the ability of language models to complete narratives containing idioms and similes. In other related research (Stowe et al., 2021), metaphors are generated based on conceptual metaphor theory (Kövecses, 2016).

While creative language generation has seen the limelight in the last few years (Lewis et al., 2021; Van de Cruys, 2020; Watanabe and Goto, 2020; Oliveira, 2017), creative language summarization has been mostly left behind. Recent endeavors in the field include 'CREATIVESUMM' (Agarwal et al., 2022): a shared task on summarization of creative documents like movie scripts, television scripts, etc. However, poems were not used in any of the sub-tasks. Other types of tasks that work with poems include poem classification (Kumar and Minz, 2014), image-to-poem generation (Xu et al., 2018), and developing better Large Language Models for poem generation through human collaboration (Chakrabarty et al., 2022b). However, work focusing on the summarization of poems is missing in prior literature. To this end, we introduce a novel poem summarization dataset.

## 3 The PoemSum Dataset

In order to expedite research in the poem summarization task, we have curated a novel dataset named 'PoemSum'. The dataset provides a diverse range of poems along with their summaries. Each sample contains the title, poet name, the poem, the source website of the poem, and finally, its corresponding summary. Appendix A.1 shows some sample poems of the dataset along with their corresponding summaries. We have provided different statistics of the PoemSum dataset in Table 1.

## 3.1 Data Collection

To construct the PoemSum dataset, we collected the poems and their corresponding summaries from different sources. While the summaries were collected from the 'Poem Analysis' website[2], which has the largest collection of high-quality poem summaries, in most cases, this website hosted only the summary but not the poem itself. Therefore, we were required to collect the poem from other sources on the internet [3]. For this reason, we collected the summaries via web scrapping, while the corresponding poems were collected manually from numerous websites.

To ensure the reliability and accuracy of the manual data collection process, four undergraduate students majoring in Computer Science and Engineering from a reputable university were hired. While selecting these individuals, it was ensured that they possessed the necessary technical skills to collect the required data (i.e., collecting the required poems from different websites).

## 3.2 Dataset Cleaning

Since the data was collected from various sources, further cleaning was required to ensure uniformity of the data. After collecting the raw data, we handled discrepancies like duplicate records, garbage values, unnecessary white spaces, corrupt values, HTML tags, and other inconsistencies. The cleaned records were further manually verified to ensure the aforementioned inconsistencies did not persist.

## 3.3 Data Verification

To verify the quality of the collected summaries, we set some standards that every sample had to maintain. The conditions are as follows:

- The summary cannot be limited to explaining only the rhyming scheme of the poem.
- The summary cannot contain only the poet's biography without any insights into the poem.
- The summary cannot only discuss other works of the poet without context.
- The summary must not only copy lines from the poem without any attempts to extract a deeper meaning.
- The summary must not be limited to rewriting the lines of the poem in simple language without any analysis.

---

| Model | R1 | R2 | RL | BS |
|---|---|---|---|---|
| T5 | **45.03** | **25.96** | **33.87** | **85.98** |
| Pegasus | 42.81 | 22.16 | 30.36 | 85.18 |
| Bart | 41.18 | 21.72 | 29.66 | 84.76 |
| mT5 | 35.01 | 14.25 | 21.91 | 83.34 |

Table 2: Performance of different Models

The summaries which did not meet these specified standards were discarded. Initially, around 3,500 samples were collected, but after an extensive data cleaning and data verification process, we ended up with 3,011 samples.

## 4 Models

We evaluated the performance of some of the standard summarization models like T5, BART, Pegasus, and mT5 on PoemSum, as they have demonstrated impressive performance in various summarization datasets in recent years (Laskar et al., 2022; Ravaut et al., 2022).

**T5:** A versatile transformer-based model (Vaswani et al., 2017) that uses a unified text-to-text transfer learning framework that can be used for diverse tasks like translation, summarization, question answering, etc. (Raffel et al., 2020).

**Pegasus:** A transformer-based model specifically designed for text generation tasks like summarization, employing pre-training with extractive supervision (Zhang et al., 2020).

**BART:** A sequence-to-sequence model based on the transformer architecture pre-trained on document denoising (Lewis et al., 2019).

**mT5:** It is an extension of T5 designed for multilingual applications. The variant we have opted to use is trained for multilingual news summarization in the XL-Sum Dataset (Hasan et al., 2021).

The pre-trained models were fine-tuned with a learning rate of 0.0001, and AdamW optimizer (Loshchilov and Hutter, 2017) with a batch size of 10. The dataset was splitted into the train, validation, and test sets following the ratio of 80:10:10.

## 5 Results and Discussions

To evaluate the model-generated summaries against the gold reference summaries, we have used the Rouge-1 (R1), Rouge-2 (R2), Rouge-L (RL), and BertScore (BS) metrics. The results are shown in Table 2. A brief description of the metrics we have used in our experiments is given below:

---

[2]https://poemanalysis.com
[3]List of sources available in our github repository

| Poem | Actual Summary | Generated Summary T5 |
|---|---|---|
| **The Death of the Ball Turret Gunner** 
 *-Randall Jarrell* 

 From my mother's sleep 
 I fell into the State, 
 And   I hunched in its belly 
 till my wet fur froze. 
 Six miles from earth, 
 loosed from its dream of life, 
 I woke to black flak 
 and the nightmare fighters. 
 When I died they   washed me out 
 of the turret with a hose. | "The Death of the Ball Turret Gunner" by Randall Jarrell is a moving poem about the death of a soldier during the Second World War. Throughout the lines of this piece, Jarrell's speaker, a deceased soldier, describes his birth and innocence while juxtaposing it with his job as a gunner and death. The first lines describe how he was born from his mother's dream into "the State" or a new, more violent understanding of life. There, he hunkers down inside a new womb, the ball turret. His days are filled with black flak and nightmarish scenes of life and death. The poem's final line reveals that the speaker is dead, and his remains were washed from the "womb" with a hose. | The Death of the Ball Turret Gunner' by Randall Jarrell describes the death of a ball-turret gunner who was shot in the head. The poem begins with the speaker stating that   he fell into the state of Alabama   and hunched in its belly     until his wet fur froze.   It was   six miles away from earth,   freed from its dream of life. When he died, it was   washed him out of the turret with a hose . |

Table 3: Sample of the Generated Summary by the T5 model. The highlighted parts show that some text is almost directly carried over from the poem, which does not help in expressing the figurative meaning of the sentences.

**Rouge (Lin, 2004):** It measures the overlap between actual and generated summary considering unigrams, bigrams, and longest common subsequence in the case of Rouge1, Rouge2, and RougeL, respectively.

**BERTScore (Zhang et al., 2019):** Instead of considering exact matches, Bertscore analyzes token similarity using contextual embeddings (Devlin et al., 2018).

The results show that the T5 model is the best-performing one, both in terms of the three types of Rouge scores and BERTScore, while the mT5 performed the worst. The key takeaway here is that the mT5 model that we use in this paper[4] was trained on a news summarization dataset. It is performing the worst as per our initial hypothesis. News data is mostly void of creative elements, which leads to an inability to express and summarize creative language even after fine-tuning. Appendix A.2 contains further explanation on this issue.

Furthermore, we have critically analyzed the results and found a few pitfalls of the evaluated models. A sample summary generated by the T5 model is mentioned in Table 3. As we can see, even in our best-performing model, the generated summary is void of the true interpretation of the poem. The actual poem summary tells us the story of a dying soldier and his final thoughts during death. The generated summary fails to carry on the ideas presented in the poem. We observe exact lines from the poem being carried over to the summary as highlighted in Table 3. It is observable that the only change in the highlighted sentences in most

cases is limited to single words like pronouns and prepositions to match the tone of the rest of the generated summary. Additionally, some unrelated words are spawned, like 'State of Alabama' instead of just 'State'. All these reveal that creative language summarization has a long way to go before it can be on par with general text summarization.

Apart from lines being carried over from the poems to the summary without plausible explanation, there are a number of other figurative language patterns like metaphors, symbolism, personification, etc., that language models are unable to comprehend (Syafitri and Marlinton, 2018; Abrams, 1999). Appendix A.2 contains a detailed insight into some of these lackings in the generated summaries.

## 6   Conclusion

In this work, we have introduced a novel dataset for poem summarization and assessed the creative language understanding capability of the popular summarization models. We have also discussed how creative language differs significantly from formal language in its structure and expression. Hence, we require different approaches when dealing with creative language, at least in the domain of summarization. After conducting several baseline experiments with the state-of-the-art summarization models, we observed that even the best-performing model was unable to produce summaries that capture the main idea of the poems in certain cases. We also used a model trained on a popular news article summarization dataset and observed that it performed worse compared to all other models we have fine-tuned. Our findings have led us to the conclusion that existing text summarization approaches and

---

[4]The mT5 model (Hasan et al., 2021) used in this work is a variant of the original mT5 model (Raffel et al., 2020)

summarization models are not well-suited for poem summarization. We believe that the proposed PoemSum dataset can serve as a valuable resource for evaluating the performance of NLP models in understanding, interpreting, and generating figurative language within the context of poetry. Meanwhile, the evaluation of Large Language Models (LLMs) (Touvron et al., 2023; OpenAI, 2023; Anil et al., 2023; Jahan et al., 2023; Laskar et al., 2023) in this task could be a good direction for future work.

## Limitations

The limitations of our work include a lower number of samples compared to other datasets for text summarization in general. The number of sources from where credible poem summaries could be collected was very limited. The poems of only one language have been included in the dataset, and hence, the dataset does not contribute toward creative language interpretation in low-resource languages.

## Ethics Statement

The poems and summaries were collected from public websites. The sources of the collection have been disclosed in our GitHub repository. At the same time, the data collectors involved in the data collection and checking process were compensated monetarily for their time and effort above the minimum wage.

## Acknowledgement

We are grateful to Islamic University of Technology (IUT) for funding the dataset curation process. We also appreciate the hard work and dedication of the data annotators and data collectors. This work would not have been possible without their support.

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

# A  Appendix

## A.1  Dataset Sample

Some sample poems and their corresponding summaries have been shown from our dataset in Table 4. Along with the poem and its summarized interpretation, the dataset also contains the poem title, poet name, and the web link to the source of the poem for each sample.

## A.2  Lackings in the Generated Summaries

We observed a number of shortcomings in the generated summaries by the evaluated models. The main issue we observed here was that the language models were having a hard time dealing with figurative language patterns. Such lackings were observed even in the case of our best-performing model, T5. The most common pattern the models seem to struggle with is metaphors, which are words or expressions whose literal usage is very different from their actual use in the poetry (Syafitri and Marlinton, 2018). The model is also seen to struggle with other patterns like personification, which means assigning of human-like qualities to any objects, and symbolism, which is an object or event that references or signifies something other than itself (Abrams, 1999).

Apart from this, another issue that was limited to the mT5 model was the generation of summaries that seemed to be inspired by new article summarization. This is completely in line with our hypothesis as the mT5 model was pretrained on news data. One of the characteristics of these summaries includes unnecessary inclusion of facts about the subject of the poem, which adds no value to the interpretation of the poem. For example, in the summary of the poem "Gold" in Table 5, the model gives us some facts about gold that are not a part of the main theme of the poem. The model also seems to be focusing too much on numbers, which would be okay for news article summarization but not in the case of poem summarization.

Table 5 shows some examples of the different lackings that we found in the case of our best-performing model T5 and also our worst-performing model mT5.

| Poem | Summary |
|---|---|
| I met a traveller from an antique land,
Who said—"Two vast and trunkless legs of stone
Stand in the desert. . . . Near them, on the sand,
Half sunk a shattered visage lies, whose frown,
And wrinkled lip, and sneer of cold command,
Tell that its sculptor well those passions read
Which yet survive, stamped on these lifeless things,
The hand that mocked them, and the heart that fed;
And on the pedestal, these words appear:
My name is Ozymandias, King of Kings;
Look on my Works, ye Mighty, and despair!
Nothing beside remains. Round the decay
Of that colossal Wreck, boundless and bare
The lone and level sands stretch far away." | "Ozymandias" by P. B. Shelley describes a traveler's reaction to the half-buried, worn-out statue of the great pharaoh, Ramses II. In this poem, the speaker describes meeting a traveler "from an antique land." The title, "Ozymandias" notifies the reader that this land is most probably Egypt since Ozymandias was what the Greeks called Ramses II. He was a great and terrible pharaoh in ancient Egypt. The traveler tells a story to the speaker. In the story, he describes visiting Egypt. There, he saw a large and intimidating statue of Ramses in the desert. He can tell that the sculptor must have known his subject well because it is obvious from the statue's face that this man was a great leader, but one who could also be very vicious. He describes his sneer as having a "cold command." Even though the leader was probably very great, it seems that the only thing that survives from his realm is this statue, which is half-buried and somewhat falling apart. |
| How good to lie a little while
And look up through the tree!
The Sky is like a kind big smile
Bent sweetly over me.
The Sunshine flickers through the lace
Of leaves above my head,
And kisses me upon the face
Like Mother, before bed.
The Wind comes stealing o'er the grass
To whisper pretty things;
And though I cannot see him pass,
I feel his careful wings.
So many gentle Friends are near
Whom one can scarcely see,
A child should never feel a fear,
Wherever he may be. | "Friends" by Abbie Farwell Brown describes the way that nature, specifically the Sky, Wind, and Sun can provide comfort if one is willing to let them. The poem begins with the speaker describing how she finds joy and peace in equal measure out in nature. She likes to go and "lie a little while" underneath a specific tree. From her spot there she sees the sky above her, looking down and protecting her. She is also able to enjoy the feeling of the dappled sunlight on her skin and take in the lace-like patterns it produces. In the second half of the poem, she describes the way the "Wind" sneaks up on her and whispers "pretty things" in her ear. Everywhere around her there is a force of nature ready to comfort her if she needs it. |
| Some day, when trees have shed their leaves
And against the morning's white
The shivering birds beneath the eaves
Have sheltered for the night,
We'll turn our faces southward, love,
Toward the summer isle
Where bamboos spire the shafted grove
And wide-mouthed orchids smile.
And we will seek the quiet hill
Where towers the cotton tree,
And leaps the laughing crystal rill,
And works the droning bee.
And we will build a cottage there
Beside an open glade,
With black-ribbed blue-bells blowing near,
And ferns that never fade. | "After the Winter" by Claude McKay is a beautiful and simple piece of poetry about an idealized fugue on the "summer isle" away from winter. The speaker spends the first stanza initiating his dream and suggesting that when winter is over, they're going to be able to "turn" their faces toward the summer isle and forget about the cold, inhospitable past. When they go towards the summer isle, they'll find peace. There will be everything that one could hope to find in the natural world. This includes rivers, flowers, cotton trees, and more. The poet uses personification to depict these scenes as well as good examples of imagery. |

Table 4: Samples of Poem and Summary from the PoemSum Dataset

| Model | Poem | Generated Summary | Lackings Observed |
|-------|------|-------------------|-------------------|
| T5 | **Bleeding Heart**
*-Carmen Giménez*

My heart is bleeding.
It bleeds upward and fills
my mouth up with salt. It bleeds
because of a city in ruins,
.......................................
Even for a bad zoning decision,
I'll bleed so much you'll be bleeding,
all of us bleeding in and out
like it's breathing,
or kissing, and because it is
righteous and terrible and red. | "Bleeding Heart" by Carmen Gimnez Smith is a powerful poem that describes how the speaker's heart is bleeding. The poem begins with the speaker stating that she has a lot of blood in her body. It is filled with salt, and it can be seen from every angle. This is something she feels when she speaks about her own life. She suffers because of everything she sees as irreproducible. In the second half of the poem, the speaker tells the reader that she is bleeding for the sake of her sister. She is also bleeding to the tips of her fingers | The generated summary interprets bleeding as actual blood flow but the poet has actually used the word "bleeding" as a **metaphor** to **symbolize** her emotional pain and sympathy. The model only considers the literal meaning of all the imagery, symbolism and metaphors used in the poem. Moreover, it directly includes some sentences from the poem in the summary. |
| T5 | **Song For a Dark Girl**
*-Langston Hughes*

Way Down South in Dixie
(Break the heart of me)
They hung my black young lover
To a cross roads tree.
Way Down South in Dixie
(Bruised body high in air)
I asked the white Lord Jesus
What was the use of prayer.
Way Down South in Dixie
(Break the heart of me)
Love is a naked shadow
On a gnarled and naked tree. | "Song For a Dark Girl" by Langston Hughes is an image-rich poem that depicts the speaker's relationship with his black young lover. The poem begins with the narrator describing how he and his white friend hung their love on trees. They hung him in Dixie, South America. He then says that he wants to break the heart of his daughter. This is not because it is going to be true. In the next lines, the poet describes what happened to him as if he was alone in the same place. | The generated summary states that the poem is about the relationship of the speaker with her lover. But the poem is actually about the tragic loss a young woman had to suffer when her lover fell victim to racial violence. The model cannot comprehend the speaker's frustration expressed through the **rhetorical question** "What was the use of prayer". The model also fails to understand the **metaphor** in the last sentence referring to the speaker's dead lover. |
| mT5 | **Gold!**
*-Thomas Hood*

Gold! Gold! Gold! Gold!
Bright and yellow, hard and cold
Molten, graven, hammered and rolled,
Heavy to get and light to hold,
Hoarded, bartered, bought and sold,
.......................................
To save - to ruin - to curse - to bless -
As even its minted coins express :
Now stamped with the
image of Queen Bess,
And now of a bloody Mary. | "Gold" by Thomas Hood is a short poem about gold. In the first lines of this poem, the speaker begins by stating that gold is an important part of the world. It is valued at more than 100m and it is used to sell for money. The second stanza focuses on the fact that Gold is one of most valuable things in the country. | The content of the generated summary is **similar to news summarization**. The generated summary only expresses the importance of gold in some factual terms but it fails to understand the contrasting nature of gold mentioned in the poem and its impact on humans. |
| mT5 | **I am water**
*-Rupi Kaur*

I am water
soft enough
to offer life
tough enough
to drown it away | "i am water" by Rupi Kaur is a short poem that speaks on the nature of water. The poem begins with the speaker stating that water is soft enough to offer life tough sufficient to drown it away. It is hard to provide life for humankind, but it is not so much as it can be done. In the next lines, the poet describes what it has to do and how it will help them to survive. | Here the model misunderstands the **personification** of water. The speaker is not actually talking about the characteristics of water. Rather, she is using the nature of water to convey that both softness and toughness can coexist within her. |

Table 5: Generated summaries by the best-performing (T5) and worst-performing (mT5) models along with criticism on their limitations