# OpenReview forum: "Unveiling the Essence of Poetry: Introducing a Comprehensive Dataset and Benchmark for Poem Summarization"
_EMNLP/2023/Conference — EMNLP 2023 Main_

### Official Review · Reviewer_dv38 · 2023-07-29

**Typos Grammar Style And Presentation Improvements:** N/A
**Soundness:** 3

**Excitement:**

2: Mediocre: This paper makes marginal contributions (vs non-contemporaneous work), so I would rather not see it in the conference.

**Missing References:**

N/A

**Paper Topic And Main Contributions:**

This paper introduces a new dataset and benchmark for poem summarization in English language.
Contributions:
- The first dataset for poem summarization called "PoemSum". It contains 3011 samples of poems in English paired with human-written summaries.
- Evaluates standard summarization models like BART, T5, Pegasus on PoemSum. Finds that they do not perform well, indicating poem summarization requires new approaches beyond text summarization.

**Questions For The Authors:**

A) It is not clear in Table 1 what is the unit of Max poem size and Max summary size, can you clarify this?
B) What figurative language patterns are present in poems that LLMs didn't capture?


**Reasons To Accept:**

- Novel task formulation: Poem summarization is a new task that requires interpreting creative, figurative language. This expands NLP research into a new domain beyond current tasks like news or scientific text summarization.
- Useful benchmark dataset: PoemSum is the first dataset for poem summarization. It can spur further research in this area and allow systematic evaluation of new models on this task. Availability of datasets is crucial for progress.

**Reasons To Reject:**

- Single language: Only English poems are included. Multilingual creative language interpretation may exhibit different challenges.
- Curation process not explained: Details of how the dataset was compiled, such as search methodology or data cleaning steps, are omitted.
- Limited analysis: The paper mostly focuses on introducing the task and dataset. Deeper analysis into model capabilities and errors is left for future work.

**Reproducibility:**

4: Could mostly reproduce the results, but there may be some variation because of sample variance or minor variations in their interpretation of the protocol or method.

**Reviewer Confidence:**

5: Positive that my evaluation is correct. I read the paper very carefully and I am very familiar with related work.

---

> ### Author Rebuttal · Authors · 2023-08-28
>
> Dear Reviewer,
>
> We extend our gratitude for the invaluable feedback you provided regarding our manuscript. We hope to address your concerns comprehensively in the subsequent text.
>
> ### Response to the questions:
>
> **Unit in Table 1:** The unit of max poem and max summary size in Table-1 is the **number of words**. This information was unintentionally not included in the submitted manuscript. We will surely add this information in the camera-ready version, along with other information like:
> | Metrics                             | Value       |
> |-------------------------------------|-------------|
> | Average **number of words** per poem    | 248  |
> | Average **number of words** per summary | 162  |
> | Average **number of poems** per poet    | 3.24        |
>
> **Regarding figurative patterns that the LLMs don’t capture:** We thank the reviewer for this critical observation. There are some figurative language patterns that are not captured by the LLMs like T5, Pegasus, Bart, mT5 we used. We agree that the inclusion of this analysis will improve the quality of our manuscript. We promise to add them in the Appendix section of our revised manuscript. Some examples are shown in the following table from our best-performing (T5) and worst-performing (mT5) models:
>
> | Model | Figurative Pattern  | Poem                                                                                                                                                                                                                                                                                                                                                                                                                                                                                                                                                                 | Generated Summary                                                                                                                                                                                                                                                                                                                                                                                                                                                                                                                                 | Lackings Observed                                                                                                                                                                                                                                                                                           |                                                            |
> |-------|---------------------|----------------------------------------------------------------------------------------------------------------------------------------------------------------------------------------------------------------------------------------------------------------------------------------------------------------------------------------------------------------------------------------------------------------------------------------------------------------------------------------------------------------------------------------------------------------------|-----------------------------------------------------------------------------------------------------------------------------------------------------------------------------------------------------------------------------------------------------------------------------------------------------------------------------------------------------------------------------------------------------------------------------------------------------------------------------------------------------------------------------------------------------------|-----------------------------------------------------------------------------------------------------------------------------------------------------------------------------------------------------------------------------------------------------------------------------------------------------------------------------|------------------------------------------------------------|
> | **mT5**    | Personification     | (i am water by Rupi Kaur) i am water soft enough to offer life tough enough to drown it away                                                                                                                                                                                                                                                                                                                                                                                                                                                                         | 'i am water' by Rupi Kaur is a short poem that speaks on the nature of water. The poem begins with the speaker stating that water is soft enough to offer life tough sufficient to drown it away. It is hard to provide life for humankind, but it is not so much as it can be done. In the next lines, the poet describes what it has to do and how it will help them to survive.                                                                                                                                                                        | Here the model **misunderstands the personification of water**. The speaker is not actually talking about the characteristics of water. Rather she is using water as a metaphor to convey that softness and toughness can coexist within her.                                                                                   |                                                            |
> | **T5**    | Metaphor            | (Ariel by Sylvia Plath) Stasis in darkness. Then the substanceless blue Pour of tor and distances  God's lioness How one we grow Pivot of heels and  knees!--The furrow ........ The dew that flies Suicidal, at one with the drive Into the red  Eye, the cauldron of morning.                                                                                                                                                                                                                                                                                       | 'Ariel' by Sylvia Plath is a beautiful poem that depicts the life of Ariel. The poem begins with the speaker stating that she is trapped in darkness. There is no substance to her body, and there is nothing to distract her from it. It is like a furrow splits and passes through its sister. She can't catch it, but it does cast shadows on her. In the second half of the poem, the speaker talks about how she has become a lioness and Godiva.                                                                                                    | The poem actually talks about the mental and emotional transformation that the rider goes through when she faces death but the prediction summary says it depicts the life of Ariel. It **fails to understand the dark metaphors**. For example- black sweet blood mouthfuls of berries indicate blood and danger in this poem. |                                                            |
> | **T5**    | Rhetorical question | (Song For a Dark Girl  by Langston Hughes)  Way Down South in Dixie (Break the heart of me) They hung my black young lover To a cross road tree.  Way Down South in Dixie (Bruised body high in air) I asked the white Lord Jesus What was the use of prayer.  Way Down South in Dixie (Break the heary of me) Love is naked shadow  On a gnarled and naked tree.                                                                                                                                                                                                    | 'Song For a Dark Girl' by Langston Hughes is an image-rich poem that depicts the speaker's relationship with his black young lover. The poem begins with the narrator describing how he and his white friend hung their love on trees. They hung him in Dixie, South America. He then says that he wants to break the heart of his daughter. This is not because it is going to be true. In the next lines, the poet describes what happened to him as if he was alone in the same place.                                                                 | The line “I asked the white Lord Jesus / What was the use of prayer” **poses a rhetorical question** that expresses the speaker’s despair and doubt about her prayer getting accepted when they face brutality because of their skin color. The model **fails to convey this** in the summary.                                                 |                                                            |
> | **T5**    | Symbolism           | (Bleeding Heart  by Carmen Gimnez Smith)  My heart is bleeding. It bleeds upward and  fills my mouth with salt. It bleeds because of a city in ruins, the chair still warm from sister's body, because it will all be irreproducible. My heart bleeds because of baby bear not finding mama bear bear and it bleeds to tips of my fingers like I painted my nails Crimson.                                                                                                                                                                                           | 'Bleeding Heart' by Carmen Gimnez Smith is a powerful poem that describes how the speaker's heart is bleeding. The poem begins with the speaker stating that she has a lot of blood in her body. It is filled with salt, and it can be seen from every angle. This is something she feels when she speaks about her own life. She suffers because of everything she sees as irreproducible. In the second half of the poem, the speaker tells the reader that she is bleeding for the sake of her sister. She is also bleeding to the tips of her fingers | The generated summary fails to capture the inner meaning of the poem. Here ‘bleeding‘ **symbolizes                                                                                                                                                                                                                            poet's sympathy and emotional pain, not actual blood flow**. |
> | **mT5**   | Contrast            | (Gold by Thomas Hood)  Gold! Gold! Gold! Gold! Bright and yellow, hard and cold Molten, graven, hammered and rolled, Heavy to get and light to hold, Hoarded, bartered, bought and sold, Stolen, borrowed, squandered, doled, Spurned by young, but hung by old To the verge of a church yard mold; Price of many a crime untold. Gold! Gold! Gold! Gold! Good or bad a thousand fold! How widely it agencies vary, To save - to ruin - to curse - to bless - As even its minted coins express : Now stamped with the image of Queen Bess, And now of a bloody Mary. | 'Gold!' by Thomas Hood is a short poem about gold. In the first lines of this poem, the speaker begins by stating that gold is an important part of the world. It is valued at more than   100m and it is used to sell for money. The second stanza focuses on the fact that Gold is one of most valuable things in the country.                                                                                                                                                                                                                          | The content of the generated summary is **similar to news summarization**. The generated summary only expresses the importance of gold but it **fails to understand the contrasting nature of gold** and its impact on humans. This example matches with our analysis in **section 5** of our paper regarding the inability of news article pretrained models when it comes to creative language summarization.                                                                                                     |                                                            |
>
> ### Response to reasons to reject:
>
> **Regarding working with a single language:** There is a scarcity of high quality poem summaries for other languages, which led us to choose only English poems as a start. It is possible to extend the work to other low resource languages with more rigorous data collection efforts. We aim to do that in our future works.
>
> **Regarding curation process:** We have discussed the data collection and cleaning process in section **3.1** and **3.2**. Due to the space limitation, we skipped a few details, which we promise to add in the final version, expanding the mentioned sections. The following text gives a detailed look into our curation process:
> 1. **Search Methodology:** We first searched for public websites with high quality English Poem Summaries. Although these were available in numerous public websites, most websites had a very small collection. Considering this, we used the website with the highest collection of summaries, as mentioned in our submitted manuscript. The summaries along with the poem name, poet name were collected via automatic web scrapping. This website hosts only the summary but not the poem itself. For the purpose of collecting the poems, four data collectors were hired and tasked to collect the poem and URL of the poem's source website given poem name and poet name. Poems that could not be found in credible websites had to be discarded from the dataset.
> 2. **Data Cleaning:** The data was first cleaned using automated techniques to remove unnecessary white spaces, garbage values and duplicate records. Sometimes html tags were included in the data due to differences in webpage structure, such inconsistencies were resolved. After automated cleaning, the data was again verified by human annotators with experience in dataset preparation.
> 3. **Data Verification:** For the purpose of data verification, another group of annotators with similar level of expertise were hired. They were tasked with cross checking the collected poems from another website other than the one which the initial collectors had listed. If any mismatch was found, the data was discarded. The overall summaries and poems were checked as well. We set some standards that every verified summary had to maintain:
> * Summary cannot be limited to explaining only the rhyming scheme of the poem
> * Summary cannot contain only the poet's biography without any insights into the poem itself
> * Summary cannot only discuss other works of the poet without context
> * Summary must not only copy lines from the poem without any attempts to extract a deeper meaning
>
>   Summaries that did not meet the specified standards were discarded.
>
> **Regarding limited analysis:** We acknowledge the importance of the identified issue for showing future research directions. In this regard, we discussed the overall performance in Table 2, and a sample use case in Table 3, to point out some of the pitfalls of an existing model. As a part of our result analysis, we observed that LLMs don't perform well in poem summarization. But models that were pretrained on news article summarization, like mT5 in our case, performed even worse. As mentioned in **section 5**, this could mean that existing summarization models cannot comprehend figurative language. Due to space constraints, we couldn't add detailed analysis in the manuscript. More analysis like the table shown above will be added in the appendix section of the manuscript.
>
> Your feedback has been instrumental in helping us identify areas that require further clarification and improvement. We hope our final manuscript will address your concerns and demonstrate the value of our work.

---

### Official Review · Reviewer_r8W8 · 2023-08-04

**Soundness:** 4

**Excitement:**

4: Strong: This paper deepens the understanding of some phenomenon or lowers the barriers to an existing research direction.

**Paper Topic And Main Contributions:**

The paper introduces "PoemSum," a novel dataset for poem summarization, aimed at understanding the figurative language of poems. It highlights the distinction between creative and formal language, revealing that current summarization models struggle to capture the main ideas of poems. The dataset is proposed as a valuable resource for evaluating language models' comprehension of figurative language in poetry, offering potential for improved understanding and generation of poetic language while advancing the field of creative language summarization. The paper presents the results of finetuning various models using this dataset vs one trained on news data only, highlighting 1) the finetunes models work better, 2) that there is still room for much improvement in the summarization of literary work.

**Reasons To Accept:**

- Summarization of literary work, and specially summarizing poems which requires a deep understanding of metaphor, symbolism, and emotion, is not an area that has been addressed enough in literature. This paper highlights this point as well as how this task differs from summarizing non-literary text and presents a good starting point for further work in this area.
- The authors have stated that the dataset described in the paper will be released to the public upon paper acceptance. Assuming that the authors release their exact splits (training, validation, and testing), the work presented will serve as a good benchmark for other researchers continuing in this direction.


**Reasons To Reject:**

- The presented dataset is some what small, but that is understandable.
- The presented results are not that impressive, but again this only highlights the difficulty of the problem being addressed

**Reproducibility:**

4: Could mostly reproduce the results, but there may be some variation because of sample variance or minor variations in their interpretation of the protocol or method.

**Reviewer Confidence:**

4: Quite sure. I tried to check the important points carefully. It's unlikely, though conceivable, that I missed something that should affect my ratings.

---

> ### Author Rebuttal · Authors · 2023-08-28
>
> Dear Reviewer,
>
> We thank you for the invaluable feedback you provided regarding our manuscript. We hope to address your concerns in the following text.
>
> ### Response to reasons to reject:
>
> **Regarding Size of Dataset:** The main issue which dictated the size was the availability of good quality poem summaries. If more summaries were available, we could have increased the dataset size even more. We hope to extend the dataset in the near future, also enriching it with poems from other languages as well.
>
> **Regarding the Obtained Results:** While the results are not impressive, we hope this work will highlight the importance and difficulty of this particular task, at the same time inspire others to come up with better architectural solutions for working with creative language.
>
> Your feedback has been helpful for reinstating the importance of the issues being addressed by our work. We hope our comments are able to properly address your concerns.

---

### Official Review · Reviewer_6WjZ · 2023-08-04

**Soundness:** 4

**Excitement:**

3: Ambivalent: It has merits (e.g., it reports state-of-the-art results, the idea is nice), but there are key weaknesses (e.g., it describes incremental work), and it can significantly benefit from another round of revision. However, I won't object to accepting it if my co-reviewers champion it.

**Paper Topic And Main Contributions:**

This paper discusses the challenges of interpreting the intended meaning of creative language in poetry and proposes a new task in natural language understanding called "Poem Summarization." The task aims to summarize poems, taking into account their deeper meanings, which are often lost if only the literal meaning is considered. The authors have created a dataset called "Poem-Sum," consisting of 3011 samples of poetry and their corresponding interpretations, which will be publicly available upon acceptance of their manuscript.

**Reasons To Accept:**

The paper's main strength lies in the innovative task it proposes and the valuable contribution through the creation of a novel dataset. The significance of interpreting figurative language in various NLG applications is highlighted effectively. The task and its associated challenges are presented and justified with great clarity. Moreover, the paper is skillfully written, making a compelling argument for the importance of this task. Additionally, it conducts a preliminary evaluation of various language models using the provided dataset.

**Reasons To Reject:**

While I recommend accepting the paper, I believe there are a few minor aspects that could enhance it. Despite being a short paper, it could include the suggestions provided below, rather than presenting an extensive and unnecessary list of source web pages.

**Reproducibility:**

2: Would be hard pressed to reproduce the results. The contribution depends on data that are simply not available outside the author's institution or consortium; not enough details are provided.

**Reviewer Confidence:**

4: Quite sure. I tried to check the important points carefully. It's unlikely, though conceivable, that I missed something that should affect my ratings.

**Typos Grammar Style And Presentation Improvements:**

The paper's corpus/data set characterization requires improvement. Enhancing it with relevant figures, possibly in Table 1, such as the average number of words/tokens per poem and summary, the time distribution of poems/authors (and potentially poems) per century, and the average number of poems per poet, would be beneficial.

---

> ### Author Rebuttal · Authors · 2023-08-28
>
> Dear Reviewer,
>
> We extend our gratitude for the invaluable feedback you provided regarding our manuscript. We hope to address the concerns you raised regarding the submitted manuscript.
>
> ### Response to reasons to reject:
> **Regarding adding minor aspects in Table-1:** We thank you for this critical observation. We agree that adding these information will improve the clarity of our manuscript. Hence the following information will be added in Table 1 of the manuscript.
> The information to be appended are as follows:
> | Metrics                             | Value       |
> |---------------------------------------|-------------|
> | Average number of words per poem | 248 |
> | Average number of words per summary | 162 |
> | Average number of poems per poet | 3.24       |
>
> We will append this information in the revised manuscripts.
>
> **Regarding time distribution of the poems:** The time distribution of poems/authors per century was not available in the sources from where the data was collected and hence we were unable to include this information.
>
> **Regarding list of webpages:** Since we could not share the dataset repository in the initial manuscript, we added a list of sources in the appendix. While we put it in the appendix to provide more transparency of the data curation process, we can also put it on the Github repo when the dataset will be released.
>
> **Regarding reproducibility:** The code and the dataset will be made publicly available along with the train, validation and test split once accepted, hence we firmly believe reproducibility will not be an issue.
>
> Your feedback has been helpful for improving vital aspects of our work. We hope our final manuscript will address your concerns and demonstrate the value of our work.

---

### Meta-Review · Area_Chair_3vBd · 2023-09-18

**Recommendation:** 4

**Metareview:**

This work proposes a new task (and a dataset) in the field of natural language understanding called "Poem Summarization".

The reviewers agree on the soundness of the proposal being good-strong, while the excitement causes more variability (mediocre-strong). Taking into account the novelty of the proposal, and that the reasons to reject have nothing to do with the excitement of the proposal, I take it being rather high. The reviewers raised reasonable and constructive comments and concerns that the authors acknowledge and will include in the final version, resulting in an improved work. The main ones being:

- Further dataset metrics
- Further analyses on figurative patterns that LLMs do not capture
- Details on curation processes

Working on a single language, should, in my opinion, do not preclude the publication of the work. Inasmuch as the code and the dataset will be made publicly available along with the train, validation and test split once accepted, I also believe reproducibility should not be an issue.

---

### Decision · Program_Chairs · 2023-10-07

**Decision:**

Accept-Main

**Comment:**

This work proposes a new task (and a dataset) in the field of natural language understanding called "Poem Summarization".

The reviewers agree on the soundness of the proposal being good-strong, while the excitement causes more variability (mediocre-strong). Taking into account the novelty of the proposal, and that the reasons to reject have nothing to do with the excitement of the proposal, I take it being rather high. The reviewers raised reasonable and constructive comments and concerns that the authors acknowledge and will include in the final version, resulting in an improved work. The main ones being:

- Further dataset metrics
- Further analyses on figurative patterns that LLMs do not capture
- Details on curation processes

Working on a single language, should, in my opinion, do not preclude the publication of the work. Inasmuch as the code and the dataset will be made publicly available along with the train, validation and test split once accepted, I also believe reproducibility should not be an issue.